# Overcoming Stability Problems in Microwave-Assisted Heterogeneous Catalytic Processes Affected by Catalyst Coking

**Ignacio Julian** [1,2,3,]*[iD], **Christoffer M. Pedersen** [4], **Kostiantyn Achkasov** [5], **Jose L. Hueso** [1,2,3], **Henrik L. Hellstern** [4], **Hugo Silva** [4], **Reyes Mallada** [1,2,3,]*[iD], **Zachary J. Davis** [4] and **Jesus Santamaria** [1,2,3]

[1] Institute of Nanocience of Aragon (INA) and Department of Chemical and Environmental Engineering (IQTMA), University of Zaragoza, 50018 Zaragoza, Spain; jlhueso@unizar.es (J.L.H.); jesus.santamaria@unizar.es (J.S.)
[2] Instituto de Ciencia de Materiales de Aragon (ICMA), Consejo Superior de Investigaciones Científicas (CSIC-Universidad de Zaragoza), 50009 Zaragoza, Spain
[3] Networking Research Centre CIBER-BBN, 28029 Madrid, Spain
[4] Center for Nano Production and Micro Analysis, Danish Technological Institute, DK-2630 Taastrup, Denmark; chm@teknologisk.dk (C.M.P.); hhel@teknologisk.dk (H.L.H.); husi@teknologisk.dk (H.S.); zjd@teknologisk.dk (Z.J.D.)
[5] Sairem, 82 rue Elisée Reclus, 69150 Décines-Charpieu, France; KACHKASOV@sairem.com
[*] Correspondence: ijulian@unizar.es (I.J.); rmallada@unizar.es (R.M.)

**Abstract:** Microwave-assisted heterogeneous catalysis (MHC) is gaining attention due to its exciting prospects related to selective catalyst heating, enhanced energy-efficiency, and partial inhibition of detrimental side gas-phase reactions. The induced temperature difference between the catalyst and the comparatively colder surrounding reactive atmosphere is pointed as the main factor of the process selectivity enhancement towards the products of interest in a number of hydrocarbon conversion processes. However, MHC is traditionally restricted to catalytic reactions in the absence of catalyst coking. As excellent MW-susceptors, carbon deposits represent an enormous drawback of the MHC technology, being main responsible of long-term process malfunctions. This work addresses the potentials and limitations of MHC for such processes affected by coking (MHCC). It also intends to evaluate the use of different catalyst and reactor configurations to overcome heating stability problems derived from the undesired coke deposits. The concept of long-term MHCC operation has been experimentally tested/applied to for the methane non-oxidative coupling reaction at 700 °C on Mo/ZSM-5@SiC structured catalysts. Preliminary process scalability tests suggest that a 6-fold power input increases the processing of methane flow by 150 times under the same controlled temperature and spatial velocity conditions. This finding paves the way for the implementation of high-capacity MHCC processes at up-scaled facilities.

**Keywords:** microwave-assisted heating; heterogeneous catalysis; catalyst coking

---

## 1. Introduction

Microwave-assisted heating has emerged as an energy-efficient heating solution for a number of processes involving chemical transformations [1–8]. In particular, microwave (MW) irradiation has exciting prospects for gas–solid heterogeneous catalysis, since the selective dielectric heating of suitable catalytic materials is capable to establish a significant gas–solid temperature gradient between the heated catalytic sample and the comparatively colder surrounding gas [9–11]. This temperature

gap is deemed responsible for the selectivity shift and the more efficient energy use reported in previous works of our laboratory. The coupled use of MW irradiation and heterogeneous catalysis to downsize the energy consumption for the manufacture of valuable products is in strong alignment with the principles of green chemistry [12–14]. The application of microwave-assisted heating to catalytic systems that operate in the absence of oxygen is, however, challenging due to the formation of coke deposits on the catalyst surface. This not only causes a fast deactivation of the catalyst but also significant stability issues. Carbon species are excellent MW-susceptors and their presence may promote the formation of hot spots and temperature gradients along the catalytic sample, which further impact both operational controllability and experimental reproducibility [10]. Under certain conditions, coke deposits may disturb the electromagnetic field within the resonator leading to cavity uncoupling and causing the instantaneous decrease of the catalytic sample temperature, thus, extinguishing the catalytic process [15,16].

The use of catalytic powders coated on structured honeycombs, made of microwave susceptor materials, takes advantage of the selective heating provided by the MW-irradiation. Essentially, MW-heating of structured catalysts promotes a significant temperature gradient between the catalyst wall and the comparatively colder gas phase within the monolith channels. The beneficial effect of the selective MW-heating on the process productivity enhancement was reported for several catalytic processes such methane [10,17,18], ethane [19] or n-hexane [20] dehydroaromatization, methane dry reforming [21], and oxidative n-butane dehydrogenation [9]. However, from an industrial standpoint, most of the reported results on MW-assisted hydrocarbon conversion refer to insufficiently long and stable reaction periods. In most cases, this is due to the troublesome management of coke formation in the presence of an intense electromagnetic field, which usually forces the MW-assisted catalytic process to a premature termination.

In this work, we explore different catalyst arrangement and reactor configuration strategies to overcome the limitations posed by coke deposition in MW-assisted heterogeneous catalytic processes. Specifically, we evaluate the thermal evolution of the dielectric properties of catalyst and catalytic support candidates for the non-oxidative methane coupling reaction (MNOC). We selected the challenging MNOC as targeting MW-assisted catalytic process, i.e. proof-of-concept process, to illustrate the role of coke deposits and how to minimize their detrimental effect on the process performance. Based on the MW-susceptibility of the selected candidates, we address the MW-heating patterns of different catalysts configurations and assess on suitable catalyst arrangements to carry out long-term MW-assisted MNOC under controlled temperature conditions.

Furthermore, we explore the scaling potentials and limitations for MW-driven heterogeneous catalytic processes affected by coking. In particular, the MW-assisted heating patterns of structured catalysts are evaluated at two different scales ($1\times$ and $150\times$), electromagnetic field modes ($TE_{111}$ and $TE_{10x}$) and MW radiation frequencies (2.45 GHz and 915 MHz) under a non-oxidative atmosphere in the presence of methane at high temperatures (> 680 °C). We aimed at providing insight into the possibilities of this technology as alternative energy-efficient solution to tackle with heterogeneous catalytic processes limited by undesired gas-phase reactions and/or coke formation at large scale.

## 2. Results and Discussion

This section provides an overview of materials dielectric characterization, experimental MW-heating tests under reactive conditions, and different reactor configurations and the technology scaling assessment.

### 2.1. Dielectric Properties of Selected Materials

The dielectric constant $\varepsilon'$ is related to the ability of a material to absorb MW radiation whereas the loss factor $\varepsilon''$ represents the ability of this material to transform the absorbed radiation into heat. Figure 1 shows the evolution of both dielectric constant and loss factor with the temperature for a number of catalytic powder samples (i.e., fresh and spent Mo/ZSM-5, carbon nanotubes) and

catalyst supports (i.e., cordierite, α- and β-SiC) participating in the MNOC process. Concerning the zeolite-based materials (Figure 1a), the mobility of the zeolite cations is the main contributor to the dielectric heating [22]. For this reason, the number and type of cations and their distribution and mobility along the zeolite cages govern the response to the electromagnetic field. On this regard, the higher the Al content, the more acidic is the zeolite and the greater is the number of exchange cations. The dielectric characterization results revealed that both dielectric constant and loss factor of the raw H-ZSM-5 (Si/Al = 11.5) are similar to these of the Mo/ZSM-5 catalyst with low metal loading (1 wt % Mo) at low temperatures (<250 °C) and lower than those reported by Nigar et al. [23] for NaY zeolites (Si/Al = 2.5, $\varepsilon'_{\text{NaY (325 °C)}} = 1.55$ and $\varepsilon''_{\text{NaY (325 °C)}} = 0.137$). This agrees with the fact that the number of exchange cations in NaY zeolites is substantially greater than that in the employed ZSM-5 (Si/Al = 11.5). The thermal evolution of the loss factor for the pristine H-ZSM-5 zeolite shows a gradual increase at temperatures above 400 °C. However, the catalysts with different Mo loadings follow a rather different pattern. The discrepancies between the dielectric properties of the pristine zeolite and the Mo-exchanged catalysts are attributed to the presence of different cationic species (H- and Mo-based cations, respectively). These cations interact differently with the zeolite support, which affects their mobility [22,24]. In particular, the number and mobility of cations in the fresh support is higher than that in the Mo-exchanged samples. The cationic molybdate species anchored at the Brønsted acid sites of the zeolite typically consists of a dimeric molybdate $[Mo_2O_5]^{2+}$, i.e. two protons are exchanged by a single Mo-based cation.

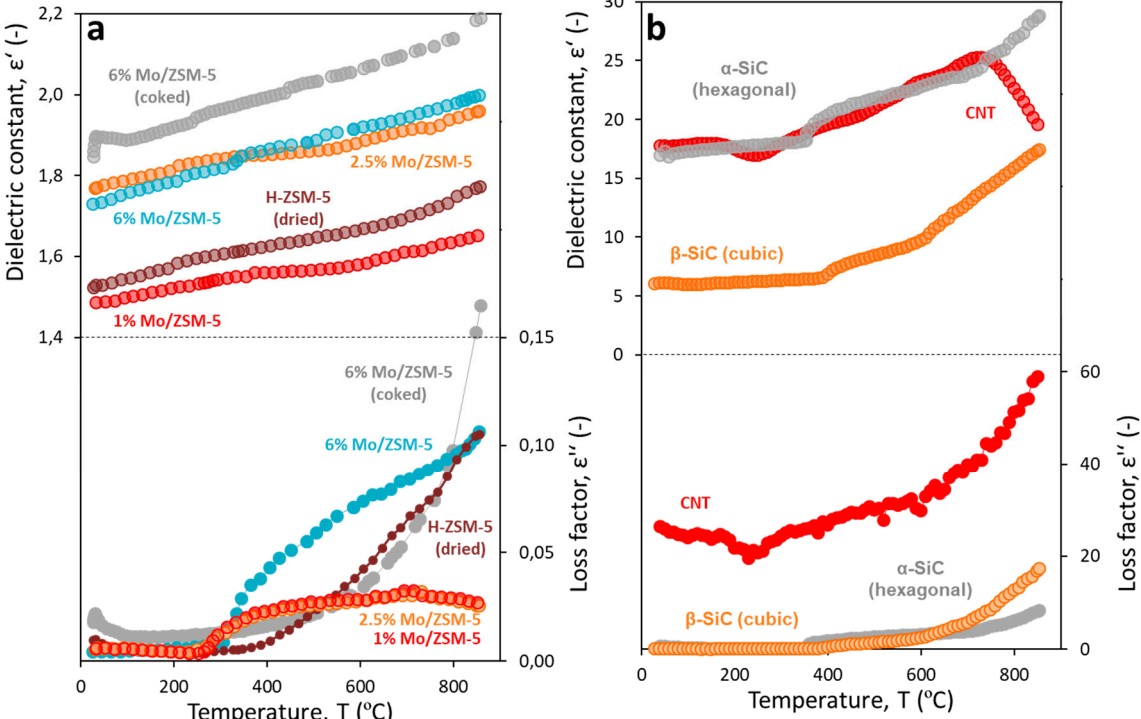

**Figure 1.** Dielectric properties of: (**a**) fresh and spent catalytic Mo/ZSM-5 powders (benchmarking catalyst for MNOC process), (**b**) carbon nanotubes (Baytubes® C 70 P) and SiC powders (100–250 μm) with different crystalline phases (cubic and hexagonal).

Regarding the effect of the metal loading, we previously observed that the higher the Mo content in the sample, the greater its MW-absorption capacity is [25,26]. The loss factor of all fresh Mo/ZSM-5 samples is very low and similar at temperatures under 300 °C, which is related to dehydration. Above this temperature, their loss factor experiences a sudden increase. This change is very significant for the sample containing 6 wt. % Mo whereas it is much softer for these containing 1 and 2.5 wt. % Mo. When zeolites are dehydrated (> 250 °C), the relaxation mechanism related to microwave heating is

linked to the mobility of extra-framework cations to different ion exchange positions [27]. To understand the different evolution of the loss factor of zeolites with different metal content, it is important to point out that the maximum Mo loading incorporated within the zeolite cages for a ZSM-5 zeolite support with the employed Si/Al ratio is around 5 wt. % This implies that the sample with a 6 wt. % Mo loading most probably contains $MoO_3$ species segregated on the catalyst surface. In contrast, the samples having 1–2.5 wt. % Mo may just contain well dispersed Mo (mono- or di-molybdate) species within the micropores of the zeolite [28–30]. Under this assumption, and considering the results shown in Figure 1a, it can be tentatively anticipated that the external Mo species may enhance the loss factor whereas the amount of well dispersed molybdates at the Brønsted sites correlate with the thermal evolution of the dielectric constant. More specifically, it appears that the molybdate species anchored at the Brønsted acid sites of the zeolite have comparatively lower mobility than that of the $MoO_3$ aggregates at the zeolite surface, which lead to higher MW-absorption capacity.

Concerning the spent 6% Mo/ZSM-5, its dielectric constant and loss factor become comparatively higher than those of the fresh catalysts at relevant MNOC temperatures (> 680 °C). As it is described elsewhere [31–34], apart from coke, the predominant Mo phases present in the spent Mo/ZSM-5 catalysts after MNOC reaction include molybdenum oxy-carbides ($MoC_xO_y$) and molybdenum carbide. Specifically, the dielectric loss of the coked 6% Mo/ZSM-5 sample at 700 °C is $\delta_{spent-700\,°C} = 7.6 \times 10^{-2}$, whereas that of the fresh one with the same metal loading is $\delta_{fresh-700\,°C} = 4.3 \times 10^{-2}$. This behavior is attributed to the excellent microwave absorption ability of the carbonaceous materials given by the displacement of the delocalized $\pi$ electrons in the presence of an electromagnetic field ($\delta_{CNT-700\,°C} = 1.59$), thus converting MW energy into heat [1,15]. Therefore, the results shown in Figure 1 suggest that the dielectric properties of the catalyst will be affected not only by changes in temperature but also by the chemical reaction.

Figure 1b shows the thermal evolution of the dielectric properties of SiC and CNT samples powder samples. As it is observed, their dielectric constant is one order of magnitude higher than that of the Mo/ZSM-5 based catalysts whereas their loss factor becomes $10^2$–$10^3$ times higher. On the contrary, cordierite is a nearly transparent material ($\varepsilon''_{cordierite\,(25\,°C)} \approx 10^{-4}$) and, thus, the thermal evolution of its dielectric properties could not be measured with the employed method [35], since heating was negligible.

These results suggest in the first place that the MW-induced preferential heating is strongly dependent on the variation of the relative dielectric properties of the selected materials (i.e., Mo/ZSM-5, cordierite, and SiC). As a result, in the event of using a Mo/ZSM-5@cordierite structured catalyst, the heating target will be the catalyst whereas in the case of Mo/ZSM-5@SiC the target will be the supporting material. The use of the cordierite-based samples will, thus, require homogeneous catalyst coating in order to get an even temperature distribution along the sample [36].

Secondly, if the catalyst coking proceeds via formation of carbon nanotubes or graphitized species, as a result of an initial low metal dispersion and presence of $MoO_3$ agglomerates at the external surface of the zeolite [25], the carbonaceous materials would absorb MW-radiation preferentially with respect to SiC supports due to their comparatively higher loss factor ($\varepsilon''$). This last scenario may result in the transient shift of the resonant frequency of the cavity and in a potentially undesired cavity uncoupling. Therefore, inhomogeneous coke deposition on the catalyst surface may lead to uneven heating and eventual hot spots formation.

## 2.2. MW-Heating Tests on Different Catalyst Configurations

MW-assisted heating tests for different structured catalyst configurations in different atmosphere, either $CH_4$ or air, were conducted in a cylindrical monomodal $TE_{111}$ MW cavity described in Section 3. These tests confirmed our previous hypothesis based on dielectric properties measurements. The combination of cordierite monoliths and Mo/ZSM-5 catalyst as MW-heating target did not allow an accurate control of the sample temperature. The sample regions having thicker catalyst coating layers became preferentially heated (Figure 2a), thus, leading to temperature distribution inhomogeneities [36].

In addition, the use of Mo/ZSM-5@cordierite configurations in the presence of reactive hydrocarbon-rich ambients under non-oxidative conditions leads to cavity uncoupling whenever coke is formed on the catalyst surface. Its preferential MW-absorption results in a significant structured catalyst temperature decrease that extinguishes the catalytic reaction immediately, i.e. MNOC in this case. For this reason, this catalyst configuration was discarded for further reaction tests.

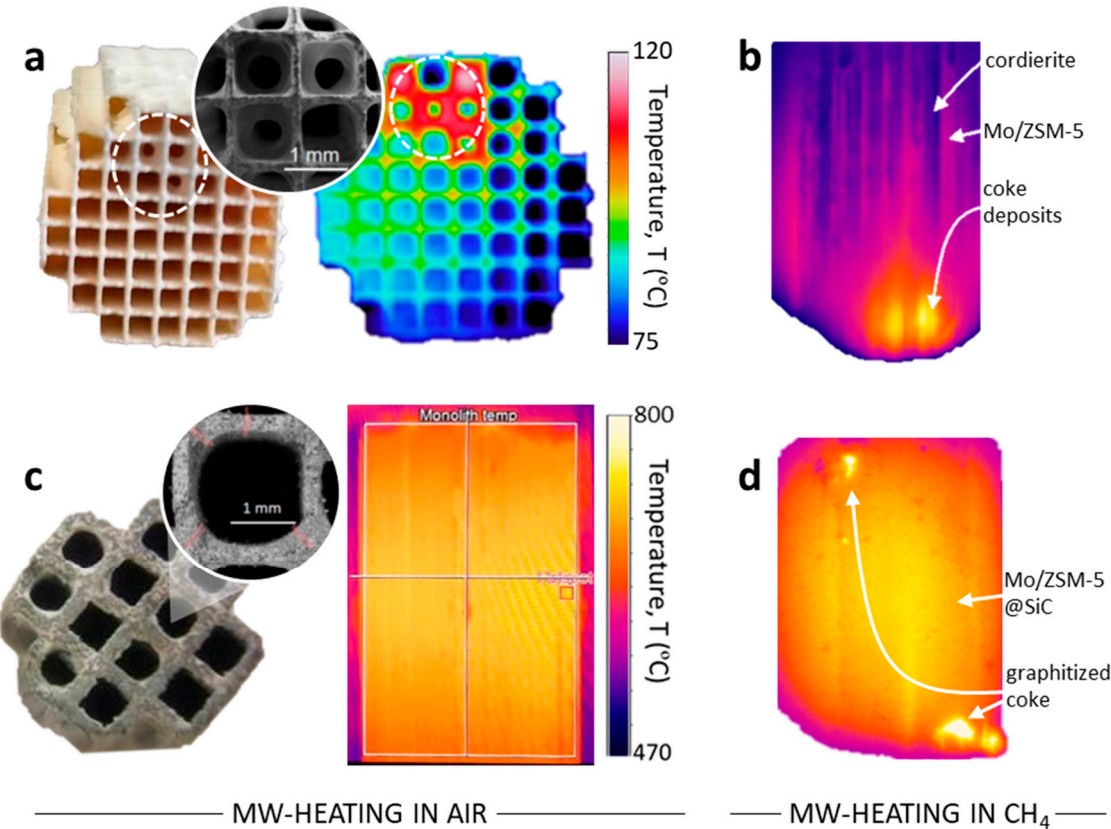

**Figure 2.** (**a**) Mo/ZSM-5 coated on cordierite monolith: coating detail and top temperature distribution under MW-heating in air; (**b**) Mo/ZSM-5@cordierite heated under $CH_4$ atmosphere: coke deposits absorb preferentially, uncouple the cavity and lead the structured catalyst to cool down; (**c**) Mo/ZSM-5 coated on β-SiC: coating detail and lateral temperature distribution under MW-heating in air; (**d**) Mo/ZSM-5@SiC heated under $CH_4$ atmosphere: hot spots formed due to coking do not disturb heating homogeneity and do not modify the average temperature along the structured catalyst. Thermal images were acquired with an infrared camera (Optris PI 1 M) pointing at the lateral wall of the MW-sample.

On the other hand, the use of β-SiC as structured support [9] for the Mo/ZSM-5 catalyst resulted in fairly homogeneous temperature distribution along the sample both in the presence of air and methane flows, regardless of the presence of unevenly coated monolith channels, i.e., catalyst accumulation at the channel corners (Figure 2c,d). The unavoidable formation of coke did not disturb the temperature profile along the Mo/ZSM-5@SiC sample and the MNOC reaction could be run for more than 5 h. This represents a promising catalyst configuration for long-term MW-assisted MNOC operation. Recently, we [10] validated this concept showing that a similar catalyst configuration based on 4% Mo/ZSM-5@SiC could perform the MNOC catalytic reaction during more than 18 hours on stream under MW-assisted heating without temperature decay or thermal runaways [10]. However, as we demonstrated, the use of Mo/ZSM-5 samples with high Mo loads (> 5 wt. % Mo) may become problematic if the initial metal dispersion is low and Mo species accumulate at the external zeolite surface. The initial carburization period in which big superficial $MoO_3$ clusters are transformed into $MoC_xO_y$ species may induce transient changes in the dielectric properties of the material. For instance,

we found that $Mo_2C$ species behave as non-dielectric materials above 200 °C (not shown). This affects the electromagnetic field distribution along the sample and makes the heating control extremely challenging (Figure 2b).

### 2.3. Reactor Configuration Assessments for Gas–Solid Catalysis in Monomodal MW-Cavities

Taking into account the above considerations, the use of β-SiC monoliths as structured supports appears as a very convenient alternative for MW-driven heterogeneous catalytic processes involving hydrocarbon chemistry [10,11]. This can be attributed to its excellent MW-absorption capacity, low thermal expansion, chemical inertness, mechanical properties, and thermal shock resistance. In addition, quartz is normally preferred as reactor tube material thanks to its very low interaction with the electromagnetic field ($\delta_{quartz\,(25\,°C)} \approx 10^{-4}$), relatively high melting temperature (> 1400 °C) and inertness. However, the choice of the structured support material and the reactor material is not the only critical issue when considering microwave-assisted catalysis. The shape and configuration of the structured material within the resonant cavity play a key role on the heating homogeneity and stability, especially for those processes affected by coking.

As a first consideration, sharp geometries such indentations should be avoided inside the cavity zone subjected to an intense EM field, since they act as electromagnetic field sinks. In the presence of methane flow at high temperatures (700 °C), highly dehydrogenated coke resembling graphite can sediment at the quartz tube, just below and above the sharp edges of the monolith in contact with the indentations and the quartz tube. The continued growth of these deposits can progressively modify the resonant frequency of the overall system (reactor + sample) and uncouple the cavity in a few minutes.

The use of quartz frits to hold the catalyst is not recommended in any case. Eventually, discrete contact points between quart and SiC monoliths may lead to the generation of hot spots, induce frit sintering, pore blocking and, eventually, explosion risks. Furthermore, the strategy of adding a spacer layer of quartz wool between the frit and the catalyst sample will lead to long-term clogging issues due to the difficulties to replace the wool. Sample holders such as tube section narrowing are also valid, as long as they are not in direct contact with the heated sample.

A suitable buffer material to be placed between the quartz tube wall and the structured SiC is fused quartz wool. This insulating material has a high melting point (around 1700 °C) and is non-combustible. Its MW-absorption capacity remains in the range of the above-mentioned aluminosilicates, slightly above pure quartz and much lower than that of SiC or CNTs. Quartz wool can, however, be heated preferentially with respect to SiC samples under MW-heating. This occurs whenever coke accumulation takes place within the porous wool, e.g. as a result of catalyst powder fall from the monolith due to adhesion issues. For this reason, it is advisable that the quartz wool, placed on both sides of the SiC monolith, is located away from the maximum EM field within the resonator.

Another typical issue is the formation of a cloudy polyaromatic condensates some centimeters above the upper-end of the monolith due to the high vertical temperature gradient between the hot sample and the surrounding (comparatively colder) atmosphere induced by MW-heating. The addition of quartz wool pieces on top of the monolith helps to maintain a relatively hot atmosphere above the catalyst outlet and to strongly decrease polyaromatics condensation. Moreover, its addition reduces the hot residence time of the gas at the outlet.

### 2.4. Scaling-Up Assessments on MW-Assisted Heterogeneous Catalytic Processes

The internationally reserved radio frequency bands for industrial, scientific, and medical use (ISM) limit the frequency bandwidth of industrially-driven microwave-assisted processes to the range 902–928 MHz, corresponding to standing wavelengths of roughly 33 cm. The comparatively broader wavelength with respect to that employed at lab-scale applications, 12.24 cm at 2.45 GHz, opens a window of process scaling by simply shifting the frequency of the MW power source. In order to assess this process scaling, we have compared the MW-heating and catalytic performance of two monomodal cavities working at different scales and with different frequency bandwidths (Figure 3).

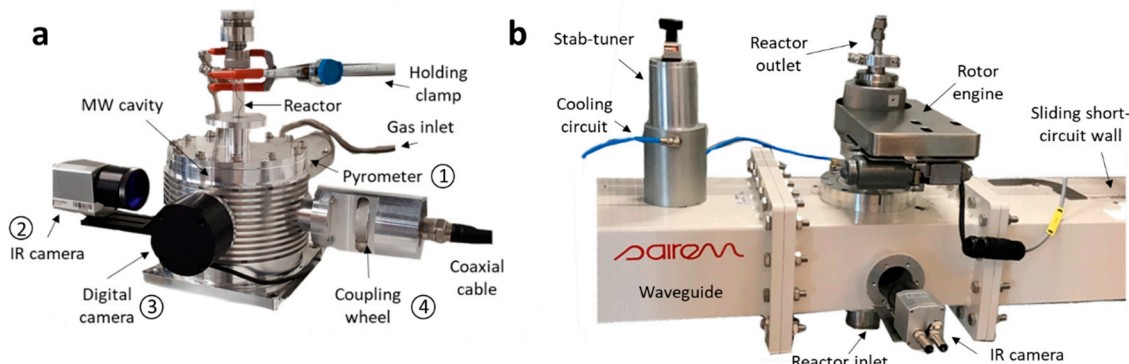

**Figure 3.** (**a**) Lateral view of the cylindrical $TE_{111}$ monomodal cavity employed for lab-scale tests, (**b**) lateral view of the scaled rectangular $TE_{10x}$ cavity employed for scaling assessment tests.

Specifically, the cylindrical lab-scale resonator uses a 110 W solid state generator in the frequency band of 2.45 GHz whereas the scaled rectangular cavity is excited by a solid state generator of 600 W (max. input power) working in the frequency band 915 ± 13 MHz.

Initially, simulations (see Section 3.5) were conducted to identify the electromagnetic (EM) field distribution along SiC monolith prototype in the lab-scale cavity. It was found that the most intense EM field appeared on both sides of the monolith, in the wave propagation direction, at their top and bottom edges (Figure 4a (number 1)). At the reactor cross-section perpendicular to the wave propagation direction, however, the EM shows a gradual increasing intensity towards the vicinity of the sample (Figure 4a (number 2)). Finally, in the region occupied by the sample, the EM field is strongly reduced due to its absorption and further transformation into heat by the monolith (Figure 4a (number 3)). In agreement with the intense EM field found at the sample edges in the simulation, MW-heating experiments under reactive atmosphere resulted in the formation of coke deposits at the quartz reactor wall along the same wave propagation axis (Figure 4a (number 4)).

The same behavior was observed for the scaled cavity. In fact, the MW-heating tests for this cavity showed a radial temperature gradient of around 210 °C towards the right side of the sample in the presence of air flow (Figure 4b). Since both reaction kinetics and catalytic performance of heterogeneous gas–solid processes strongly depend on the temperature of the catalyst, such a temperature gradient does not allow controlling any catalytic process in the scaled cavity. This limitation becomes even more evident for such processes dominated by coking. A straightforward solution to minimize the radial temperature gradient was to provide rotation to the reactor vessel containing the heated sample, as it is done for standard multimodal cavities. For this purpose, gas-tight swivel fittings were placed at both reactor ends and a rotor device was attached to the external reactor wall. The rotation of the sample at moderate velocities (4–6 rpm) provided steady-state uniform temperature distribution along the scaled monolith (Figure 4b). The maximum radial temperature gaps along the outer wall of the monolith were below 30 °C, working at a mean temperature of 700 °C. At these conditions, the maximum temperature gradient along the vertical axis was below 100 °C.

The combination of Mo/ZSM-5@SiC material, reactor rotation, sample-edge polishing, and quartz wool arrangement drastically improved the scaled MW-heating performance. However, under certain reactant flow regimes, an accumulation of catalyst particles at the glass wool was observed. The detachment of some catalytic particles from the SiC monolith surface, probably due to attrition and adhesion issues, can potentially result in glass wool coking and cavity uncoupling. Therefore, in order to perform long-term catalytic tests at the scaled MW-MNOC set-up without heating performance decay by means of coke accumulation and cavity uncoupling, a solution based on the enlargement of the monolith length was adopted.

The use of longer monoliths (length: 12 cm) allowed both upper and lower ends of the sample to be away from the most intense EM field, thus, preventing coke formation at such ends. Using this

reactor configuration, the MW-assisted MNOC process could be run at 640 °C for more than 43 h on stream without apparent cavity uncoupling. Furthermore, under non-reacting conditions, it was found that the scaled system was able to keep average sample temperatures around 700 °C using air flows up to 15 $L_N$/min with a power input of 600 W (and without detecting any reflected power back to the MW source). Summarizing the previous results in terms of scalability, the bigger MW set-up allowed heating 150 times greater catalyst loadings and, thus, has the potential to process 150 times higher reactant flows (working at the same spatial velocity) by just using a 6-fold microwave power input with respect to the lab-scale $TE_{111}$ cavity. This finding opens a new promising scenario for the implementation of high-capacity MW-assisted heterogeneous catalytic processes affected by detrimental gas-phase parallel reactions and/or coke generation.

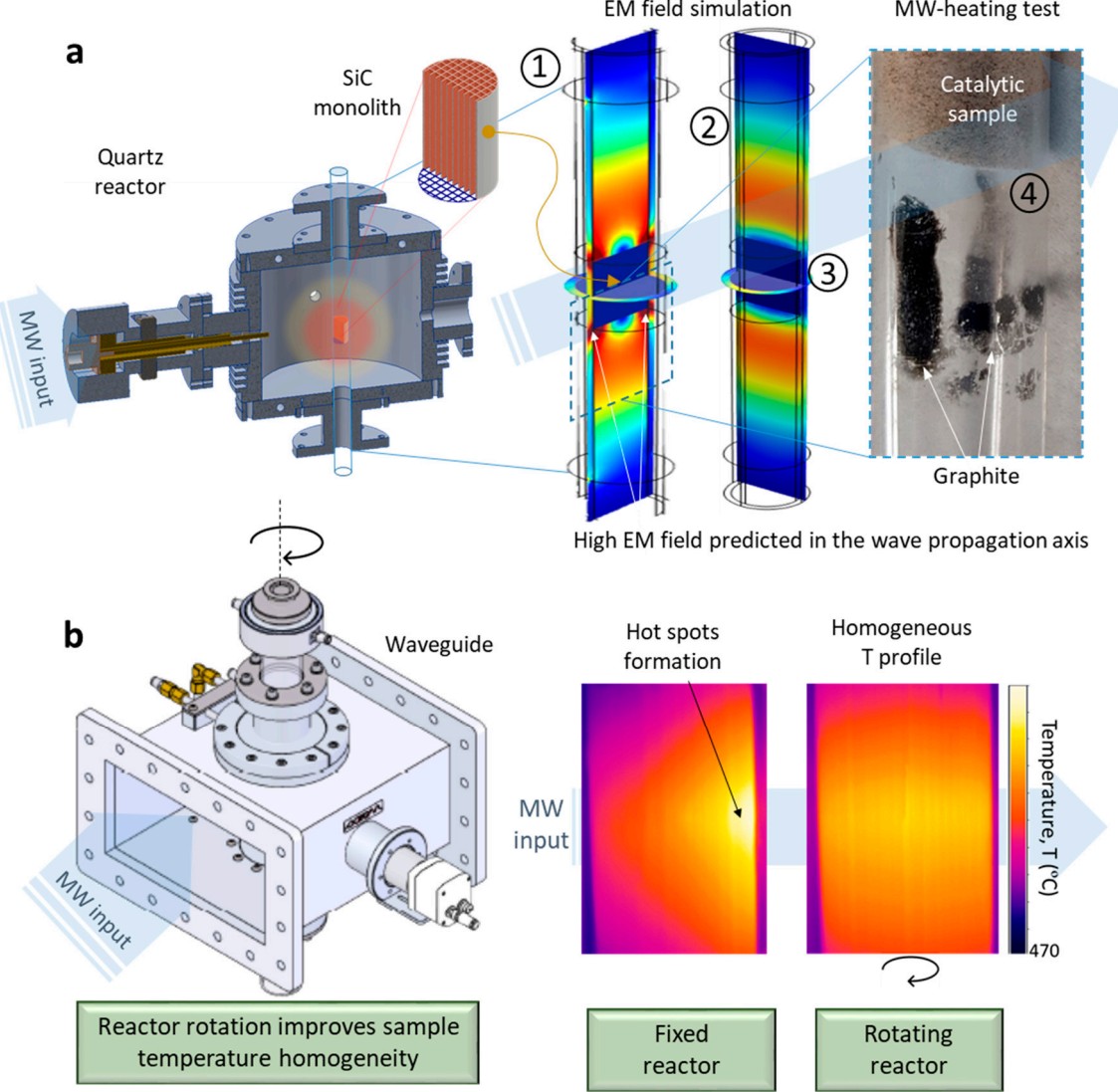

**Figure 4.** (**a**) Scheme of the lab-scale cylindrical cavity; simulated longitudinal and transverse EM field distribution maps at the inner reactor region; experimental coke deposition below the catalytic sample under MW-assisted MNOC operation; (**b**) scheme of the rectangular cavity with detail of reactor rotation; temperature distribution map along the scaled monolith wall for standing and rotating configurations.

## 3. Materials and Methods

### 3.1. Lab-Scale MW-Cavity Set-Up

The lab-scale monomodal cavity consists in a stainless-steel cylindrically-shaped resonator working in a $TE_{111}$ resonant mode, in which the maximum electric field is found at the cavity center with a substantially constant field area (Figure 3a). This microwave resonator prototype was designed and fabricated by the DIMAS group (Polytechnical University of Valencia, Spain). The cavity diameter and height are 105 mm and 85 mm, respectively. The theoretical resonant frequency for that mode was calculated to be 2.482 GHz. The exact frequency of operation will depend on the dielectric properties of the heated material and, thus, will change dynamically throughout the heating process, as the dielectric properties generally change with temperature. The maximum power input provided by the solid-state MW generator (RCA2026U50, RFcore Ltd. Gyeonggi-do, South Korea) is 110 W. The hollow guide sections at the top and bottom of the cavity prevent MW from escaping from the cavity and keep all energy confined within. The quartz tube containing the catalytic sample is located across the hollow guide, being the sample placed at the applicator center. The selected sample containers are quartz tubes due to their transparency to microwave radiation (moderate dielectric constant and very low loss factor) and high temperature resistance (up to about 1200 °C). Both the position and dimensions of the material were selected to ensure uniform heating of the sample volume using a commercial electromagnetic simulator (QWED-3D) [35]. The maximum sample size was defined as a cylinder being diameter = 9.5 mm and height = 15 mm. In order to allow the flow of reactive gases through the sample, the solution required a quartz tube with a quartz frit or preferably quartz wool, as porous support, to hold the sample in the tube center.

Four 8 mm diameter access ports were designed in the side wall of the cavity in addition to the ports at the top and bottom for the quartz tube immersion: 1) local temperature inspection via pyrometer; 2) inspection of temperature profile along the sample via IR camera; 3) visual inspection via digital video camera; 4) scrollable excitation monopole probe for applicator tuning (N-type coaxial connector). The dimensions and positions of the access holes in the cavity were designed in order to minimize the electric field disturbance, while keeping the emissions at levels below the safety limits. The penetration depth of the monopole into the cavity can be mechanically adjusted with a moving wheel. The experimental set-up is illustrated in Figure 3a.

The temperature of the sample was measured by an infrared (IR) camera and a pyrometer pointing at the sample through the 8 mm access ports. The relative position of each non-contact thermometer with respect to the sample and the microwaves inlet port is illustrated in Figure 4a. The pyrometer (Optris CTlaser LT) works in a temperature range from −50 °C to 975 °C using a spectral range of 8–14 μm and indicates the temperature of the external quartz wall of the reactor. In contrast, the IR camera (Optris PI 1 M) works in a 0.92–1.1 μm spectral range that does not interfere with quartz transmission. This allows the direct measurement of the sample wall temperature in the range 450–1800 °C. Temperature corrections were applied in order to estimate the sample temperature as a function of its emissivity at each temperature. The resonator prototype incorporates PID control software that is able to tune the MW frequency bandwidth at a given input power, in order to control the sample heating rate and target temperature, when required. The user sets the targeted temperature slope and the software auto-tuning adapts the frequency bandwidth in which MW are emitted (0.1–100 MHz around 2450 MHz). PID and fuzzy controls use the output signal of infrared cameras and pyrometers pointing to the heated sample to control the average temperature on the external wall. Additionally, the penetration depth of the monopole probe can be adapted to meet heating requirements in case the dielectric properties of a material change dramatically as its temperature increases.

### 3.2. Scaled MW-Cavity Set-Up

The scaled monomodal cavity consists of a rectangular resonator working in a $TE_{10x}$ resonant mode, for which the maximum electromagnetic field can be tuned along the waveguide using a three

stub-tuner and a sliding short-circuit (Figure 3b). The resonator prototype was designed and fabricated by Sairem. The applicator dimensions were designed so that the working mode is close to the typical frequency of industrial applications (915 MHz). The resonator is operated by a solid-state MW generator with maximum power output of 600 W (GLS600W, Sairem, Décines-Charpieu, France). The solid-state generator is equipped with a frequency scan and auto-tune functions [37] that can be employed to find the resonant frequency at every heating scenario and cavity configuration in order to minimize the reflected power. The chimneys at the top and bottom of the cavity prevent MW from leaking and keep all the energy confined within the cavity. A set of contactless thermometers are installed to monitor the sample temperature. 1) An infrared pyrometer (Micro epsilon CTM-3SF75H1-C3, T = (150–900 °C)) is threaded to the rear cavity hole. A signal conditioner is employed to display and monitor the temperature of the quartz wall in contact with the monolith sample. 2) A thermal IR camera (Optris PI 1 M, T = (450–1800 °C)) is placed at the front flange pointing to the central region of the cavity through the dedicated chimney located at the cavity center. The low spectral range of this camera allows determining the temperature of the sample wall directly, being transparent to quartz. The 45 mm o.d. quartz reactor (41 mm i.d.) is introduced through the top and bottom chimneys positioned at the center of the resonator. A custom-designed water-cooled vacuum connector with Viton O-ring was used as gas-tight solution for the reactor inlet. The employed SiC monoliths have an external diameter of 4 cm, a total length of 12 cm and an overall mesh size of 400 cells per squared-inch (cpsi).

Experimental heating tests were conducted using different cavity coupling scenarios, trying to maximize both the average monolith temperature and the temperature distribution homogeneity. Since the main purpose was to evaluate the system capacity to reach MNOC operational temperatures (T = 700 °C) throughout the monolith, both the maximum MW source power (600 W) and realistic vertical gas throughputs (0.3–20 L/min compressed air) were employed in all cases.

### 3.3. Structured Catalysts Preparation

An incipient wetness impregnation method was employed to incorporate the molybdenum precursor into the parent H-ZSM-5 zeolite (Zeolyst CBV2314, $SiO_2/Al_2O_3$ = 23). The required amount of commercial ammonium heptamolybdate, $(NH_3)_6Mo_7O_{24} \cdot 4H_2O$ (Sigma-Aldrich, St. Gallen, Switzerland) was dissolved in deionized water and poured dropwise into the zeolite under continuous mixing. Catalysts with the following Mo loadings were prepared: 1, 2.5, 4, and 6 wt. % Mo. The impregnation step was followed by drying and calcination in air at 550 °C for 6 hours [38].

For the experimental tests at the lab-scale MW-cavity, two types of structured monoliths were employed, cordierite ceramic (2 $MgO$:2 $Al_2O_3$:5 $SiO_2$, Corning, New York, United States) and silicon carbide (β−SiC, SICAT), being their cell density 400 and 200 cells-per-squared-inch (cpsi), respectively. The parent monoliths were cut down into small cylindrically-shaped samples of 15 mm height and 9.5 mm diameter to be tested in the monomodal MW resonator. This specific monolith size was selected to overlap with the maximum electromagnetic field generated within the MW cavity. The scaled SiC monoliths were employed as purchased (Landson Emission Technologies A/S, Svendborg, Denmark). Prior to the washcoating process, the outer surfaces of the monoliths were covered with Teflon tape to avoid coating on the external walls. The Mo/ZSM-5 catalyst was deposited on the surface of the monoliths following a conventional wash-coating method. This process involved the following steps: a) monolith immersion in the slurry solution containing the desired material, b) monolith withdrawal at constant speed, c) removal of the slurry excess from the channels, d) solvent evaporation, and e) calcination. The reagents used to prepare the slurry were deionized water, PVA (polyvinyl alcohol 85,000–124,000 MW, Sigma-Aldrich) as binder, colloidal silica (LUDOX AS-30, Sigma-Aldrich) to increase the viscosity of the slurry and improve the adherence to the support and the zeolite-based catalytic powder (Mo/ZSM-5). The following molar composition was employed, ($H_2O$: PVA: $SiO_2$: Mo/ZSM-5) = (61: 3: 6: 30), adapted from a previous protocol reported by Eleta et al. [39]. The resulting slurry viscosity was 0.17 Pa·s. An axial withdrawal velocity of 10 mm/min was employed for all coating tests [10].

The synthesis of materials has been performed by the Platform of Production of Biomaterials and Nanoparticles of the NANOBIOSIS ICTS, more specifically by the Nanoparticle Synthesis Unit of the CIBER in BioEngineering, Biomaterials & Nanomedicine (CIBER-BBN).

### 3.4. Dielectric Properties Measurements

In order to evaluate the catalysts and supporting materials suitability for MNOC under microwave irradiation, the dielectric permittivity ($\varepsilon$) (dielectric constant and loss factor) of different materials was measured. The complex permittivity value ($\varepsilon = \varepsilon' + j\,\varepsilon''$) depends on the dielectric constant ($\varepsilon'$) and the loss factor ($\varepsilon''$). The loss tangent (tan $\delta = \varepsilon''/\varepsilon'$) gives a measure on the dissipation of energy in the form of heat whereas the dielectric constant ($\varepsilon'$) is related with the capacity of the material to absorb the MW radiation.

For each test, a packed bed of each powder sample was loaded into a quartz reactor tube (height: 300 mm; external diameter: 12 mm). The employed bed size was the same in all cases, being its dimensions: 15 mm height and 9.5 mm diameter. All samples were preheated in air (360 °C) in order to remove water from the powder. The dielectric characterization of the powders was carried out using the so-called cavity perturbation method in a dual mode cylindrical cavity following the methodologies described by Catalá-Civera et al. [35]. Specifically, the cavity cell was designed to have two dominant modes ($TE_{111}$ and $TM_{010}$) employed for sample heating and dielectric properties measuring, respectively. These two electromagnetic fields were induced by two probes placed at the lateral wall (N-type probe) and bottom-side of the resonator (SMA probe), respectively. The $TM_{010}$ was selected as measuring mode because it causes a larger resonant frequency shift and, thus, provides higher sensitivity for the dielectric measurements. The theoretical resonant frequencies of both modes for the given cavity dimensions are 2.432 GHz and 2.187 GHz. The use of a cross-coupling filter with a rejection level greater than 60 dB at the heating band and low insertion losses at the testing band (<3 dB) allowed avoiding cross-coupling between heating and testing operations.

The set of samples includes: Mo/ZSM-5 ($SiO_2/Al_2O_3$ = 23, Zeolyst CBV2314) catalysts with different metal loads and as received multi-walled carbon nanotubes (CNT, Baytubes® C 70 P), $\alpha$- and $\beta$-SiC powders (SiCAT), and cordierite (Corning). Mo/ZSM-5 was selected as the benchmarking MNOC catalyst for the production of olefins and aromatics. Cordierite and SiC are standard inert catalyst supports in structured arrangements and carbon nanotubes are well known highly MW-absorptive materials that, in addition, may be generated as by-product of the MNOC process at the catalyst surface.

### 3.5. Simulation of the Electromagnetic Field Distribution in Monomodal Cavities

The commercial simulation software package, COMSOL 5.1, was used to model the electromagnetic field, power absorption, and sample temperature in the $T_{111}$ and $T_{10\times}$ cavities. Three different physics were considered in the model: electromagnetic waves, heat transfer and fluid dynamics [23]. For simplicity, the simulated sample consisted in an uncoated SiC monolith. A physics-controlled mesh size was adopted. The magnetic field contribution was neglected, the variation of the dielectric properties of each material with temperature was defined by the user and the 3D wave equation was only applied in the air domain inside the waveguide. In particular, the experimental dielectric values obtained for $\beta$-SiC were employed. The thermal conductivity and heat capacity of the materials were considered temperature independent, the heat flux by radiation was neglected and the convective velocity field around the reactor was simulated with fluid dynamics to account for natural convection. A 'free flow' package was adopted to model the gas flow through the quartz reactor and the solid sample channels under no-slip conditions.

## 4. Conclusions

This work assessed the use of MW-assisted heating equipment for heterogeneous catalytic process affected by coking. The evolution of the dielectric properties with temperature of different catalyst and catalytic support candidates for the non-oxidative coupling of methane (proof-of-concept

reaction) was evaluated. Suitable catalyst configuration and reactor arrangements were proposed to counteract the detrimental effect of coking on the MW cavity uncoupling and to extend the catalytic process operation. The use of catalyst coatings on SiC monoliths enabled working for several hours under reacting atmospheres keeping a constant reaction temperature even in the presence of coke deposits. The scalability of MW-assisted heterogeneous catalytic processes was evaluated working at two different MW radiation frequencies and cavity sizes. It was found that MW energy input and sample size are not linearly correlated: a 6× power increase allowed processing 150× greater reactant flows. This opens a new promising scenario for the implementation of high-capacity MW-assisted heterogeneous catalytic processes and, in particular, of those affected by detrimental gas-phase parallel reactions and coke generation.

**Author Contributions:** This study was conducted through contributions of all authors. Conceptualization J.S., R.M., and J.L.H.; Methodology I.J. and R.M.; Formal analysis I.J., C.M.P., and R.M.; Funding acquisition J.S. and Z.J.D.; Resources J.S., Z.J.D., and K.A.; Investigation I.J., C.M.P., H.S., H.L.H., J.L.H., and K.A.; Supervision J.S., Z.J.D., and R.M.; Writing—review and editing I.J., J.L.H., and R.M.

**Funding:** This research was funded by the European Union's Horizon 2020 Research and Innovation Programme (ADREM project–Grant Agreement no. 680777) and the APC have been waived by the journal.

**Acknowledgments:** The synthesis of materials has been performed by the Platform of Production of Biomaterials and Nanoparticles of the NANOBIOSIS ICTS, more specifically by the Nanoparticle Synthesis Unit of the CIBER in BioEngineering, Biomaterials & Nanomedicine (CIBER-BBN).

**Conflicts of Interest:** The authors declare no conflict of interest.

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
