# Peer review of "Overcoming Stability Problems in Microwave-Assisted Heterogeneous Catalytic Processes Affected by Catalyst Coking"

_catalysts, doi:10.3390/catal9100867_

Round 1

Reviewer 1 Report

This manuscript assessed the use of MW-assisted heating equipment for heterogeneous catalytic  process affected by coking. The authors was to investigate the evolution of the dielectric properties with temperature of different catalyst and catalytic support candidates for the non-oxidative coupling of methane. The manuscript is well constructed and the idea of the work seems to be very interesting. The results are clearly presented, and the conclusions are consistent with the results obtained. However, the authors, in the introduction section, should value more the advantages of the use of a heterogeneous catalyst in green chemistry [Thomas, J. M.; Raja, R.; Lewis, D. W. Single-site heterogeneous catalysts. Angew. Chem. Int. Ed. 2005, 44, 6456. DOI:10.1002/ANIE.200462473; Procopio, A.; Cravotto, G.; Oliverio, M.; Costanzo, P.; Nardi, M.; Paonessa,. R. An Eco-Sustainable Erbium(III)-Catalysed Method for Formation/Cleavage of O-tert-butoxy carbonates. Green Chem., 2011, 13, 436-443. DOI: 10.1039/C0GC00728E] and the study of the dependence on time dielectric heating of heterogeneous catalyst [Procopio, A.; De Luca, G.; Nardi, M.; Oliverio, M.; Paonessa, R. General MW-assisted grafting of MCM-41: Study of the dependence on time dielectric heating and solvent. Green Chem., 2009, 11, 770–773. DOI: 10.1039/B820417A].

The authors discuss an example of heterogeneous catalysis MW assisted. The authors presented the work in a detailed way and well represented by the figures used. The results and discussions are well exposed as well as the conclusions. The description of the spectrometer used for the measurements of dielectric properties is missing. for the rest I believe that the manuscript is good for the reproduction in catalyst.

The manuscript can be accepted after a minor revision.

Reviewer 2 Report

The article is devoted to overcoming the limitations caused by coke deposition in MW-assisted heterogeneous catalytic processes. The authors showed the evaluation the use of different catalysts and reactor configurations. Authors proposed use of catalyst coatings on SiC monoliths, that improved stability of the process. Another finding was that microwave energy input and sample size are not linearly correlated.

The article is well-written, all findings are clearly presented. I’m not a specialist in English, but in my opinion the language is correct.

In my personal opinion, the scientific value of the article is not high – the article can be interesting only for a narrow group of industry specialists, rather than scientists. The article can be published with a few corrections, listed below.

The authors did not describe experimental setup used for measurements of dielectric properties: what kind of spectrometer was used? It does not follow from the text. The authors should supplement the text with this information. Figure 5 cannot be included in the serious scientific text. It is the picture of the laboratory, not the professional illustration of the experimental setup. I suggest just to remove this pix.
